# Sex-dependent effects of *Setd1a* haploinsufficiency on development and adult behaviour

**Matthew L. Bosworth**[1], **Anthony R. Isles**[1], **Lawrence S. Wilkinson**[1,2,3], **Trevor Humby**[1,2,3]*

**1** Division of Psychological Medicine and Clinical Neuroscience, MRC Centre for Neuropsychiatric Genetics and Genomics, School of Medicine, Cardiff University, Cardiff, United Kingdom, **2** School of Psychology, Cardiff University, Cardiff, United Kingdom, **3** Neuroscience and Mental Health Research Institute, Cardiff University, Cardiff, United Kingdom

* HumbyT@cardiff.ac.uk

**Data Availability Statement:** All data are publicly available at the following 'Open Science Framework' link: https://osf.io/gf3d7/.

**Funding:** This work was supported by a Wellcome Trust Integrative Neuroscience PhD grant (109084/

## Abstract

Loss of function (LoF) mutations affecting the histone methyl transferase *SETD1A* are implicated in the aetiology of a range of neurodevelopmental disorders including schizophrenia. We examined indices of development and adult behaviour in a mouse model of *Setd1a* haploinsufficiency, revealing a complex pattern of sex-related differences spanning the pre- and post-natal period. Specifically, male *Setd1a*⁺/⁻ mice had smaller placentae at E11.5 and females at E18.5 without any apparent changes in foetal size. In contrast, young male *Setd1a*⁺/⁻ mice had lower body weight and showed enhanced growth, leading to equivalent weights by adulthood. Embryonic whole brain RNA-seq analysis revealed expression changes that were significantly enriched for mitochondria-related genes in *Setd1a*⁺/⁻ samples. In adulthood, we found enhanced acoustic startle responding in male *Setd1a*⁺/⁻ mice which was insentitive to the effects of risperidone, but not haloperidol, both commonly used antipsychotic drugs. We also observed reduced pre-pulse inhibition of acoustic startle, a schizophrenia-relevant phenotype, in both male and female *Setd1a*⁺/⁻ mice which could not be rescued by either drug. In the open field and elevated plus maze tests of anxiety, *Setd1a* haplosufficiency led to more anxiogenic behaviour in both sexes, whereas there were no differences in general motoric ability and memory. Thus, we find evidence for changes in a number of phenotypes which strengthen the support for the use of *Setd1a* haploinsufficient mice as a model for the biological basis of schizophrenia. Furthermore, our data point towards possible underpinning neural and developmental mechanisms that may be subtly different between the sexes.

## Introduction

Epigenetic modifications controlling chromatin structure and organisation, can enhance or suppress gene expression dependent on the mechanism in action. Methylation of lysine residue 4 on histone 3 (H3K4), generally considered an activating histone mark [1], is catalysed by

Z/15/Z) and an A. Bruce Naylor Memorial Early Career Research Fellowship from The Waterloo Foundation awarded to MLB; UKRI Medical Research Council (MRC) IMPC: Pump Priming Award (MR/P026176/1) awarded to ARI, LSW and TH; ARI, TH and LSW are members of the MRC Centre for Neuropsychiatric Genetics and Genomics (MR/L010305/1). For the purpose of Open Access, the author has applied a CC BY public copyright licence to any Author Accepted Manuscript version arising from this submission. The funders had no role in study design, data collection and analysis, decision to publish, or preparation of the manuscript.

**Competing interests:** The authors have declared that no competing interests exist.

a histone methyltransferase complex, of which *SETD1A* (SET Domain Containing 1A) is an essential subunit [2]. *SETD1A* has been implicated in a range of biological functions, including cell cycle regulation [3–5], maintenance of pluripotency in embryonic stem cells [6–8] and neuronal progenitors [9, 10], and DNA repair [4, 11, 12], highlighting a wide distribution across different cell types, throughout different stages of development and during adulthood [2]. Exome sequencing studies have implicated loss of function (LoF) mutations in the *SETD1A* gene that increase susceptibility for a number of neurodevelopmental disorders (NDD) and conditions, including developmental delay, early-onset epilepsy, intellectual disability and schizophrenia [13–16]. While these genetic variants are rare (occurring in 0.13% of schizophrenia cases), they are highly penetrant and their effects on gene function (i.e. haploinsufficiency) can be recapitulated in model systems. Consequently, *SETD1A* provides a biologically tractable target for disease modelling.

Homozygous knock-out of *Setd1a* is embryonically lethal[8]. Consequently, non-human models have targeted haploinsufficiency (*Setd1a*[+/-]), with current mouse models employing either frameshift mutations to exon 7 or 15 or disruption to exon 4 via an upstream LacZ/Neo cassette [17–20] to investigate the mechanisms of Setd1a function and how it might influence NDD. Haploinsufficiency also recapitulates the clinical LoF mutations found in human conditions for more direct translational impact. From a neurobiological perspective, changes including axonal branching deficits, impaired synaptic plasticity, regionally-specific gene expression changes linked to morphology and dendritic complexity, synapse formation, and abnormal cortical ensemble activity have been observed [17–21]. *Setd1a* may also have a role in regulating the balance between neuronal progenitor cells (NPC) proliferation and differentiation in neurogenesis [8–10], suggesting that perturbations to this balance may play a role in pathogenic mechanisms of *Setd1a* LoF in neurodevelopmental disorders. Behaviour in these models has also been affected, with impairments in working and spatial memory, sensorimotor gating, hyperactivity, and altered juvenile social behaviour [17–20]. Other work in *Drosophila* has shown that short- and long-term courtship memory is impaired by conditional knockdown of Set1 in neurons of the mushroom body [22].

Setd1a may have important roles in early development, as shown in the maternal high-fat diet model in mice, a manipulation that is associated with schizophrenia-relevant phenotypes in animal models [23, 24], where *Setd1a* expression is reduced in the placenta [24]. There is further evidence that Setd1a can influence placental function and development from studies in *Setd1a* haploinsufficient mice via its actions on embryonic stem cell (ESC) pluripotency and differentiation [6–8], with some evidence of gestational compromise though no differences in growth to adulthood [17]. Based on these findings, it might be predicted that *SETD1A* LoF causes disturbed neurodevelopment with downstream consequences for later brain function that contribute to the emergence of atypical patterns of behaviour and psychopathology, possibly via effects on placental function.

Therefore, using a newly created mouse model of Setd1a haploinsufficiency, where Setd1a is disrupted in exon 4 [25], we investigate how deletion of Setd1a may influence pre- and post-natal growth and behaviours in adulthood. In this model, we have previously demonstrated downregulation in genes enriched for mitochondrial pathways in frontal cortex from E14 to P70, but no differential enrichment in genes associated with schizophrenia [25], here we replicate these results at a similar time point (E13.5). We also find evidence for sex-linked developmental changes that extend into early post-natal growth, with effects on male and female embryo:placenta ratios that suggest imbalance in demand and delivery capacity at different foetal ages, and in and male post-weaning catch-up growth. In terms of adult behaviour, we found evidence for an enhanced acoustic startle response in male Setd1a[+/-] mice and increased anxiety-related behaviour in both male and female Setd1a[+/-] mice. Prepulse inhibition of the

acoustic startle response was attenuated in both sexes. However, the sensorimotor gating deficits could not be rescued by haloperidol or risperidone, suggesting that these antipsychotic agents are ineffective for ameliorating schizophrenia-relevant phenotypes in *Setd1a*$^{+/-}$ mice. Taken together our findings strengthen the support for the use of *Setd1a* haploinsufficient mice as a model for the biological basis of schizophrenia and point towards possible underpinning neural mechanisms, that may be sexually dimorphic.

## Materials and methods

### Animals

All procedures were conducted in accordance with the UK Animals (Scientific Procedures) Act 1986. Animal studies and breeding were approved by the Cardiff University's Ethical Committee and performed under a United Kingdom Home Office project license (PP1850831, ARI).

*Setd1a*$^{+/-}$ mice were produced using a strain created by the Knockout Mouse Phenotyping Consortium. To generate a model with a constitutive germline transmissible knockout allele, male C57BL/6NTac-Setd1a$^{tm1c(EUCOMM)Wtsi}$/WtsiCnrm mice (obtained from MRC Harwell) were paired with female B6.C-Tg(CMV-cre)1Cgn/J mice (obtained from The Jackson Laboratory) [25]. F1 male progeny were genotyped to identify animals heterozygous for the Setd1a$^{CMV-cre/+}$ allele. These were crossed with C57BL/6J females (obtained from Charles River) to enable removal of the X-linked CMV-cre transgene from males in the F2 generation. Experimental cohorts were generated by pairing male F2 *Setd1a*$^{+/-}$ mice with C57BL/6J females (wildtype, WT). These mice show haploinsufficiency as demonstrated by a 48.8% reduction in mRNA at E13.5 in *Setd1a*$^{+/-}$ brain compared to WT, with comparable reductions in levels of SETD1A protein (46.3% reduction) (S1 Fig). Animals for the assessment of post-natal growth (N = 84) and behavioural studies (N = 77 and N = 56) were housed in mixed-genotype cages with littermates of the same sex (2–5 per cage) in a vivarium maintained on a 12-hour light-dark cycle (lights on from 08:00–20:00) at a temperature of 21 (± 2)°C and 50 (± 10) % humidity. Experiments were conducted during the light phase. Standard laboratory chow and water was available *ad libitum* throughout all experiments, and mice were carefully monitored for signs of ill health throughout the studies.

### RNA-Seq for schizophrenia common variants

RNA was extracted from 16 (8 WT and 8 KO, balanced for sex) E13.5 whole brains using a Direct-zolTM RNA Miniprep kit (Zymo, UK). E13.5 tissue was selected based on the results of the developmental expression work, which showed highest expression levels at this timepoint, and to supplement the data in Clifton et al (2022) [25]. Detailed methods can be found in S2 Fig. In brief, following sample purification, concentration and integrity assessment Library preparations were performed according to manufacturer's instructions (KAPA mRNA Hyperprep kit. Roche, Switzerland) with 1 µg total RNA as input. Library concentrations were adjusted prior to pooling to achieve a final DNA molarity of 20 nM and sequencing was performed on a HiSeq 4000 (Illumina, USA) with 75 bp paired-end reads at a read depth of 30 million reads. Reads were trimmed to remove adapters and low-quality bases, and passed FastQC quality control [26], before mapping to the mouse reference genome (GRCm38) using STAR (mean number of reads mapped was 97.2%, SD = 0.5%). Read counts, generated using featureCounts were allocated to genomic features using the mouse Ensembl gene annotation (GRCm.38.95). Differential expression analysis was performed using DEseq2 [27] implemented in R (version 3.5.3). The Benjamini Hochberg correction was used to correct for

multiple testing and identify genes that were differentially expressed in *Setd1a+/-* relative to WT (*p*adj < .05), with sex included as a covariate.

GO term enrichment analysis was performed using the Database for Annotation, Visualization and Integrated Discovery [28]. Only protein-coding genes with entrez IDs (219 genes) were included in the differentially expressed gene-set. A custom background gene-set was created using all expressed genes in the dataset (12,295 genes). This was defined as any gene with a FPKM > 1 across 8 samples (i.e., the number of biological replicates in each condition). The differentially expressed gene-set was filtered to include only genes with human entrez IDs (205 genes). As previously, with the hypothesised links between *Setd1a* and risk for schizophrenia, a custom background gene-set was assessed for enrichment of schizophrenia common variants using MAGMA [29] with GWAS summary data from 40,675 cases and 65,643 controls [30]. The differentially expressed gene-set was filtered to include only genes with human entrez IDs (205 genes), and gene-set SNPs were filtered to include those with a minor allele frequency of less than 0.01.

## Assessment of embryo and placental weights, placental gene expression and postnatal growth

For the neurodevelopmental investigations, timed-matings were conducted between C57BL/6J x C57BL/6NTac parents and pregnant dams were culled by cervical dislocation. Tissue was obtained from at least two separate litters per timepoint. Placental and embryo wet weights were recorded at E11.5, E13.5 and E18.5 and an embryo:placenta weight ratio was calculated by dividing embryo wet weight by placental wet weight (see S1 Table for litter sizes and N). Whole placentas and embryo tissue biopsies were snap frozen using dry ice and stored at -80°C until required. Genotyping was conducted using DNA extracted from the tissue biopsy to identify $Setd1a^{+/-}$ mice and determine embryonic sex. Animals for the assessment of postnatal growth (males: WT, N = 23, $Setd1a^{+/-}$ mice, N = 18 and females: WT, N = 25, $Setd1a^{+/-}$ mice, N = 18) were weaned on postnatal day 28 and weighed at maternal separation (P28) and then every seventh day thereafter until P70. Thus, growth curves were assessed after maternal separation to eliminate potential effects of early-life stress induced by repeated handling on behaviour in adulthood.

## Behavioural testing

Two cohorts of mice were used to assess the effects of *Setd1a* haplosufficiency on adult behavior. For each study, run orders were counterbalanced across animals (although male mice always preceded females), with the experimenter blinded to the genotype of the subjects. The first cohort (Cohort 1: males: WT, N = 21, $Setd1a^{+/-}$ mice, N = 18 and females: WT, N = 20, $Setd1a^{+/-}$ mice, N = 18), a subsample of the animals used to evaluate postnatal growth, were tested in adulthood (aged 2–3 months at start of testing in the following order: elevated-plus maze (EPM), open field test (OFT), locomotor activity, sensorimotor gating, rotarod test, and novel object recognition. The second cohort of mice (Cohort 2: males: WT, N = 16, $Setd1a^{+/-}$ mice, N = 10 and females: WT, N = 17, $Setd1a^{+/-}$ mice, N = 13), was first used to test whether the effects of *Setd1a* haploinsufficiency on ASR and PPI observed in the first cohort could be replicated (S3 Fig). These mice were then randomly divided into two approximately equal groups (balanced for sex and genotype, see Fig 3 for n) and allocated to either the haloperidol or risperidone condition, to test the effects of these antipsychotics on sensorimotor gating impairments. All apparatus was cleaned with 70% (v/v) ethanol between animals, and testing was conducted under dimmed lighting (15 lux) except for locomotor activity which was run in the dark.

## Sensorimotor gating

The acoustic startle response (ASR) and prepulse inhibition (PPI) were measured using apparatus from SR-Lab (San Diego Instruments, USA). Animals were placed in a clear Perspex tube (35 mm internal diameter) mounted on a Perspex plinth in a sound-attenuating chamber. A 70 dB (A scale) white noise stimulus was continuously played throughout the session via a loudspeaker positioned 120 mm above the tube. The whole-body startle response was detected on each trial by a piezoelectric sensor attached to the plinth, which transduced flexion in the plinth into a digitised signal. The average startle response (Vavg) was recorded in arbitrary startle units using SR-Lab software over the 65 ms period following stimulus onset. Startle data were weight-adjusted by dividing Vavg by body weight recorded immediately after the test session. PPI was calculated as the percentage reduction in startle amplitude between prepulse and pulse-alone trials (excluding the first three pulse-alone trials). Each session of 96 trials, started with a five-minute habituation period, and was divided into three blocks. Pulse amplitude was set to 120 dB and 105 dB in block 1 and block 2, respectively and the third block comprised a range of pulse-alone trials (80–120 dB in 10 dB increments), with three of each trial type. Acoustic stimuli were presented with a mean intertrial interval of 16 seconds (pseudorandomly varied between 6 and 24 seconds). Each pulse-alone trial consisted of a 40 ms 120db or 105db startle stimulus. Prepulse trials consisted of a 20 ms prepulse stimulus followed by a 40 ms startle stimulus 80 ms after prepulse offset. In the first two blocks, 6 consecutive pulse-alone trials were presented followed by 7 additional pulse-alone trials interspersed with 18 prepulse trials (either 4, 8 or 16 dB (above background) with six trials of each prepulse amplitude). Only the data from 120 dB pulse alone and the 8 and 16 dB prepulse trials are reported here for brevity, but all raw data are available in the online repository. For the pharmacological investigations using the second cohort of mice, with haloperidol and risperidone, a reduced session protocol was used, whereby all 4 dB prepulse trials and all trials with 105 dB stimuli were removed.

## Anxiety-related behaviour

The open field test (OFT) and elevated-plus maze (EPM) were used to assess anxiety-related behaviour under identical lighting conditions (15 lux) and the position of each mouse in the apparatus was tracked using EthoVision XT software (Noldus Information Technology, Netherlands) at a rate of 12 frames/s via a camera mounted above the centre of each piece of apparatus. The OFT comprised a 750 x 750 mm arena with 45 cm white Perspex walls. Each animal was placed in the same corner of the arena and allowed to explore freely for 10 minutes. The arena was divided into two, concentric virtual zones: the 'inner zone' (central 600 x 600 mm) and the 'outer zone' (surrounding 150 mm). The EPM was constructed of white Perspex and consisted of four arms of equal size (175 x 78 mm) extending from a central square region (78 x 78 mm) and positioned 450 mm above the floor. Two of the arms were 'open' (no walls) and two were enclosed by 150 mm high opaque walls. Arms of the same type were diametrically opposed. Each animal was placed in the same enclosed arm at the start of the trial and allowed to freely explore the apparatus for five minutes. Measures of anxiety were the patterns of exploration between the different arms or zones of the apparatus, with particular focus on the time in the central zone of the OFT and the latency to enter and time on the open arms of the EPM. Other measures included total distance moved, zone or arm transitions and the numbers of head dips and stretch attend postures made by the mice on the EPM.

## Locomotor activity

Locomotor activity levels were assessed using previous methods [31], with the mice placed into clear Perspex chambers (210 x 360 x 200 mm) with two transverse infrared beams positioned 30 mm from either end of the chamber and 10 mm above the floor of the chamber. The

apparatus was linked to a computer using ARACHNID software (Cambridge Cognition Ltd., UK). Activity levels were recorded as the number of beam breaks during each session using a custom programme (BBC BASIC Version 6). Each animal completed three 2 hours sessions over consecutive days at the same time of day.

## Rotarod test

Motor learning and co-ordination were assessed using a mouse Rotarod (Model 47600, Ugo Basile, Italy). The rod (30 mm diameter) was coated with rubber grooves to provide grip. Each animal completed five trials across two days (three trials on day one and two trials the next day). During each 5 minutes trial, the speed of rotation increased from 5–50 rpm at a constant rate of 0.15 rpm/s. The latency to fall was recorded on each trial, when a mouse caused a timer to stop when triggered by a lever (160 mm below the rod). On trials where the animal stayed on the rod for the duration of the trial, the latency to fall was recorded as the maximum trial length.

## Novel object recognition

Recognition memory was assessed using the novel object recognition (NOR) paradigm [32] with retention intervals of 30 minutes and 24 hours. All mice completed the experiment at both the 30 minute and 24 hours retention interval, with order counterbalanced across animals and at least 24 hours between each condition. For the second test session, new objects that had not been encountered previously were placed in the quadrants of the arena that were unoccupied in the previous session. Location of the objects and allocation of objects to retention interval were counterbalanced across animals, with the experimenter blinded to the genotype of the subjects. Testing was conducted in a white Perspex arena (300 x 300 mm) with 300 mm high walls, and objects were common household items (bottles, cans, containers) of approximately the same size (120, x 50 mm). Prior to testing, mice were habituated to the empty arena for 10 minutes per day over three consecutive days. Each test session comprised three phases, conducted over one of two days as appropriate, and between phases mice were returned to a holding cage in the test room. The 'habituation phase', where the mice explored the empty arena for 10 minutes, preceded the 'acquisition phase', in which mice were returned to the arena and allowed to explore two identical objects (in diagonally opposite quadrants of the arena, 105 mm from the corner) for up to 15 minutes. To control for potential differences in object neophobia that could contribute to subsequent memory performance, the total amount of object exploration (defined as when the animal's head was within 20 mm and oriented towards the object) was timed during the trial by the experimenter. The acquisition phase was ended once 40 seconds of object exploration was achieved or the maximum trial length had elapsed. Animals were then returned to either a holding cage or their home cage for a retention interval of 30 minutes or 24 hours. In the 'test phase', mice were placed back in the arena with one of the objects that they had been exposed to during the acquisition phase ('familiar') and another object that had not been encountered previously ('novel'). The objects were presented in the same locations as the acquisition phase, and the mice allowed to explore freely for five minutes, and the amount of familiar and novel object exploration was recorded manually by the experimenter using EthoVision XT software (Noldus Information Technology, Netherlands). The main measure of novel object recognition was a discrimination ratio (d1) calculated from the object contact times for each test session, with the following equation: novel/novel+familiar.

## Drug preparation and administration

Haloperidol (Sigma, UK) and risperidone (Sigma, UK) were initially dissolved in 1M glacial acetic acid and then diluted in 0.9% saline such that an equal volume (100 μL per gram of

bodyweight) was administered for each injection. Drugs were administered at two doses (0.5 mg/kg and 1.0 mg/kg, i.p.), with a vehicle control treatment, with a 30 minutes pre-treatment delay before testing. This regimen and these doses were selected based previous studies [33, 34] and pilot dose-response data showing that a 0.5 mg/kg dose of either haloperidol or risperidone did not have significant effects on sensorimotor gating in WT C57BL/6J mice (S4 Fig). At 1.0 mg/kg, haloperidol caused a significant increase in PPI and risperidone caused a significant reduction in the ASR. Therefore, 0.5 mg/kg was selected as a sub-threshold dose to explore drug effects in $Setd1a^{+/-}$ mice in the absence of non-specific effects of the drugs on sensorimotor gating that occur in WT mice. Mice from Cohort 2 were divided into two groups for the assessment of risperidone or haloperidol, respectively. Dose order was counterbalanced across animals for vehicle and 0.5 mg/kg and there was a seven day washout between sessions. All mice received the 1.0 mg/kg dose in the final session to explore drug effects at a dose that was known to affect sensorimotor gating in WT animals to rule out the possibility that the lack of effect in $Setd1a^{+/-}$ mice was due to an insufficient dose of drug.

## Statistical analysis

Data were analysed using IBM SPSS software (version 25). Placental and embryo weight data were analysed using 3-way MANOVA with between-subject factors of GENOTYPE (WT and $Setd1a^{+/-}$), SEX (male and female) and AGE (E11.5, E13.5, E18.5) and a covariate of litter size. Simple effects were analysed by 1-way ANOVAs with a between-subject factor of GROUP (male WT, male $Setd1a^{+/-}$, female WT, female $Setd1a^{+/-}$). Analysis of postnatal growth included the within-subjects factor of AGE (7 levels) and the between-subject factors of GENOTYPE and SEX. ASR data were analysed using ANOVA with GENOTYPE and SEX as between-subjects factors, and PPI data were analysed with PPI-INTENSITY (8 dB and 16 dB) as an additional within-subjects factor. Effects of haloperidol and risperidone were analysed separately by ANOVA with GENOTYPE and SEX as between-subjects factors and a within-subjects factor of DOSE (vehicle, 0.5 mg/kg and 1.0 mg/kg). Data from the EPM and OFT were analysed by MANOVA with between-subject factors of GENOTYPE and SEX, and additional within-subjects factors of ZONE (central, outer) or ARM (open, closed) were included to assess exploration across the OFT or EPM, respectively. Data from one female WT animal was excluded from analysis of the OFT because it was identified as a multivariate outlier. Locomotor activity, rotarod and NOR data were analysed using ANOVAs with between-subject factors of GENOTYPE and SEX and the additional within-subjects factors of DAY (3 levels), TRIAL (5 levels) or DELAY (30 minutes and 24 hours), respectively. Data from one male WT subject was missing from the NOR due to a technical issue during testing. Greenhouse Geisser corrected results are reported for locomotor activity and the rotarod test because a significant Mauchly's test result indicated that the assumption of sphericity had been violated. Bonferroni-corrected *post-hoc* tests were used to probe significant effects where appropriate. Data are shown as mean±SEM, and criterion level of significance was set at the 0.05 level.

## Results

### Sex-related effects of *Setd1a* haplosufficiency on placental weight and gene expression, and post-natal growth

Analysis of embryonic development through gestation, at time points where *Setd1a* mRNA expression and protein levels remained relatively constant (S5 Fig). Litter size remained relatively consistent at each of the gestational ages assessed, with litters showing Mendelian ratios of WT and $Setd1a^{+/-}$ embryos and equal numbers of male and female embryos (S2 Table).

Male *Setd1a*$^{+/-}$ embryos were generally lighter than WT (~6%), and female *Setd1a*$^{+/-}$ embryos slightly heavier than WT (~4%), however these differences were not significant (Fig 1A, main effect of GENOTYPE, $F_{1,105} = 0.32$, $p = 0.57$), at any timepoints investigated (GENOTYPE*-AGE, $F_{2,105} = 0.61$, $p = 0.94$). Male embryos weighed more than female (main effect of SEX, $F_{1,105} = 6.54$, $p = 0.01$), but genotype did not interact with sex to influence embryo weight (GENOTYPE*SEX, $F_{1,105} = 3.43$, $p = 0.07$). As expected, embryos increased in weight throughout gestation (main effect of AGE, $F_{2,105} = 5358.26$, $p<0.001$), but growth was not affected by

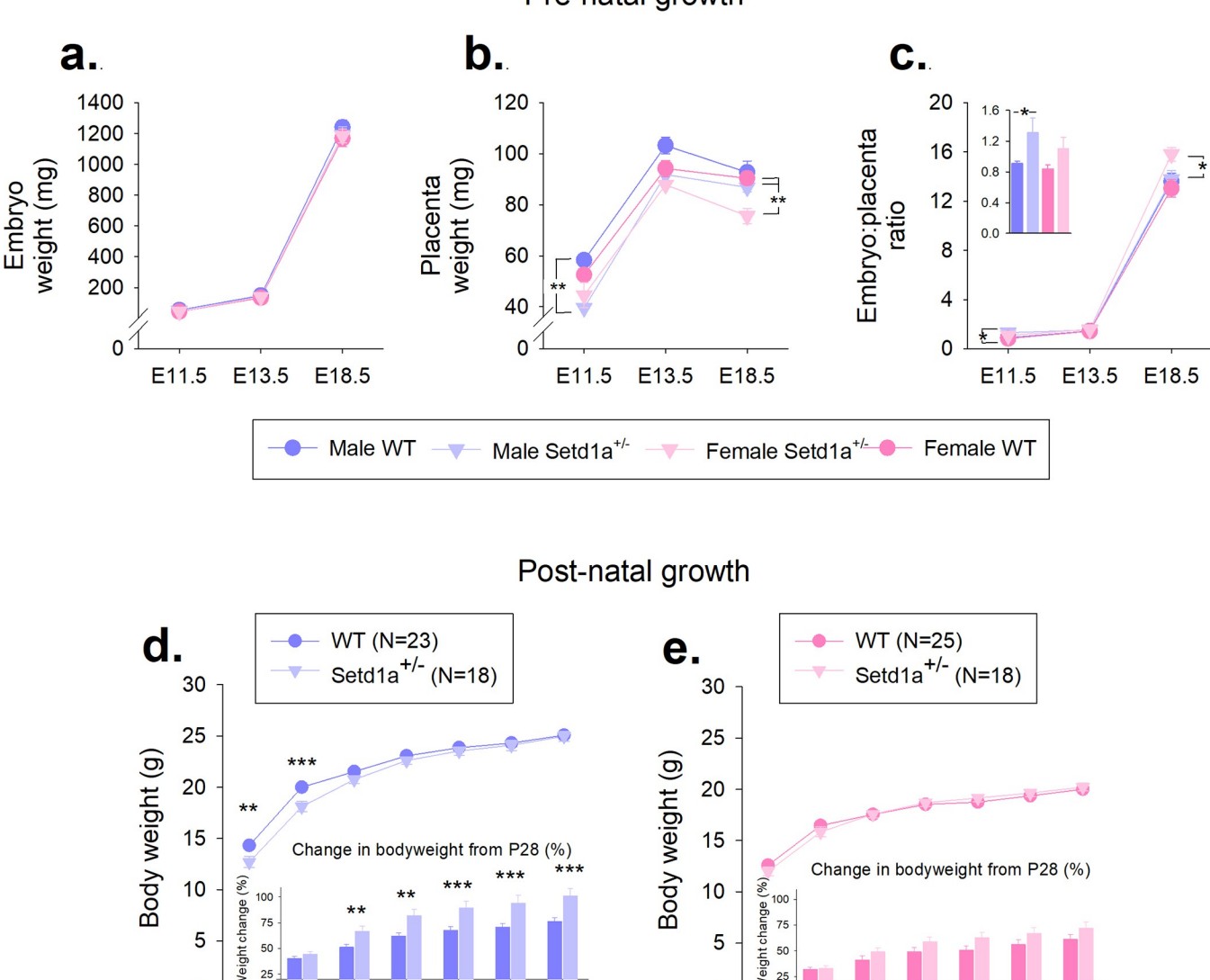

**Fig 1. Sex-related effects of *Setd1a* haplosufficiency on placental size, gene expression, and post-natal growth.** WT and *Setd1a*$^{+/-}$ embryo (a) and placental (b) weights, and embryo:placenta ratios (c) from three pre-natal time points (Inset bar graph shows E11.5 data). Post-natal growth in male (d) and female (e) WT and *Setd1a*$^{+/-}$ mice between P28 until P70. Inset bar graphs (d and d) show proportional increase in weight from P28. N for a-c, see Supplementary Methods Table 1 for litter sizes and N *, ** and *** show significant comparisons between WT and *Setd1a*$^{+/-}$ at $p<0.05$, $p<0.01$ and $p<0.001$, respectively. Data shows mean±SEM.

genotype or sex (3-way interaction, $F_{2,105}$ = 0.40, p = 0.67). A significant 3-way interaction demonstrated complex effects of AGE and SEX on the size of placentae from $Setd1a^{+/-}$ and WT embryos (Fig 1B, $F_{2,105}$ = 6.21, p = 0.003). At E11.5, simple effects analysis showed a significant effect of GENOTYPE ($F_{3,23}$ = 7.56, p = 0.001) with *post-hoc* testing highlighting a significant reduction (~32% smaller) in weight between male $Setd1a^{+/-}$ and male WT placentae (p<0.001). At E18.5, placentae from female $Setd1a^{+/-}$ placentae were significantly reduced in size (~16% smaller) compared with both male and female WT (main effect of GENOTYPE, $F_{3,39}$ = 7.38, p = 0.001 and *post hoc* testing p<0.003). Placentae were generally smaller in $Setd1a^{+/-}$ embryos than WT, across all time points evaluated (main effect of GENOTYPE, $F_{1,105}$ = 37.11, p<0.001). These placental weight differences were reflected by increased embryo:placenta ratios (EPR) in $Setd1a^{+/-}$ embryos relative to WT (Fig 1C, main effect of GENOTYPE, $F_{1,105}$ = 16.98, p<0.001). Further analysis indicated elevated EPR in male $Setd1a^{+/-}$ embryos relative to male WT at E11.5 (main effect of GENOTYPE, $F_{3,23}$ = 3.98, p = 0.02 and *post hoc* testing p = 0.05) and female $Setd1a^{+/-}$ embryos at E18.5 (main effect of GENOTYPE, $F_{3,39}$ = 3.25, p = 0.03 and *post hoc* testing p = 0.04). As embryo weight increased with embryonic age, the EPR also increased, as placenta size plateaued (main effect of AGE, $F_{2,105}$ = 2266.42, p<0.001). Litter size, as a covariate, impacted on embryo size ($F_{1,105}$ = 10.20, p = 0.02), but did not affect either placenta size ($F_{1,105}$ = 0.61, p = 0.44) or EPRs ($F_{1,105}$ = 3.13, p = 0.08).

$Setd1a$ haploinsufficiency, in this model, did not cause embryonic lethality as demonstrated by WT:$Setd1a^{+/-}$ ratios at birth which did not deviate from Mendelian ratios ($X^2_{10}$ = 1.39, p>0.05), and median litter size across 13 litters was 7 (S2 Table), which is equivalent to reported sizes for C57BL/6J mice [35]. Proportions of males and females was also as expected ($X^2_{10}$ = 1.39, p>0.05). Furthermore, all animals appeared in good general health and low pre-weaning mortality was observed (2.3%). Assessment of body weight (Fig 1D and 1E) from the point at which pups were separated from dams (P28) revealed that, as expected male mice of both genotypes were heavier than females, with a significant increase over the first few weeks post-weaning (AGE*SEX interaction, $F_{2.26,180.93}$ = 63.79, *p*<0.001). A significant interaction was found between GENOTYPE and AGE ($F_{2.26,180.93}$ = 11.43, p<0.001) and *post-hoc* analysis revealed that male $Setd1a^{+/-}$ mice were significantly lighter than their WT littermates at maternal separation (P28; p = 0.005), an effect that was sustained one week later (P35, p<0.001). By P42, male $Setd1a^{+/-}$ and WT mice were of equivalent body weight (*p* = 0.06), indicating that catch-up growth had occurred. This was confirmed by analysing the percentage increase in body weight from P28 (Fig 1D and 1E, inset bar charts). Specifically, body weights of $Setd1a^{+/-}$ male (but not female) increased significantly more than their WT littermates from P42 onwards, the age at which there was no longer a genotype effect on body weight.

## RNA-Seq highlights mitochondrial differences, but not for schizophrenia common variants in E13.5 $Setd1a^{+/-}$ brain

267 genes (Benjamini Hochberg $p_{adj}$<0.05) were differentially expressed (234 downregulated and 33 upregulated) in brains of $Setd1a^{+/-}$ mice at E13.5 (Fig 2A). Modest changes in gene expression were observed, with log2 fold changes ranging from -0.63 to 1.19. Hierarchical clustering based on normalised read counts was performed using Morpheus (https://software. broadinstitute.org/morpheus) to produce a dendrogram (Fig 2B). This showed that samples of the same genotype clustered together, suggesting similar patterns of gene expression changes within each group, which were taken forward for the *Gene ontology* term enrichment analysis.

As shown in Table 1, results of the *Gene ontology* term enrichment analysis revealed 14 nominally significant (uncorrected *p*<0 .05) associations, but only the mitochondrion

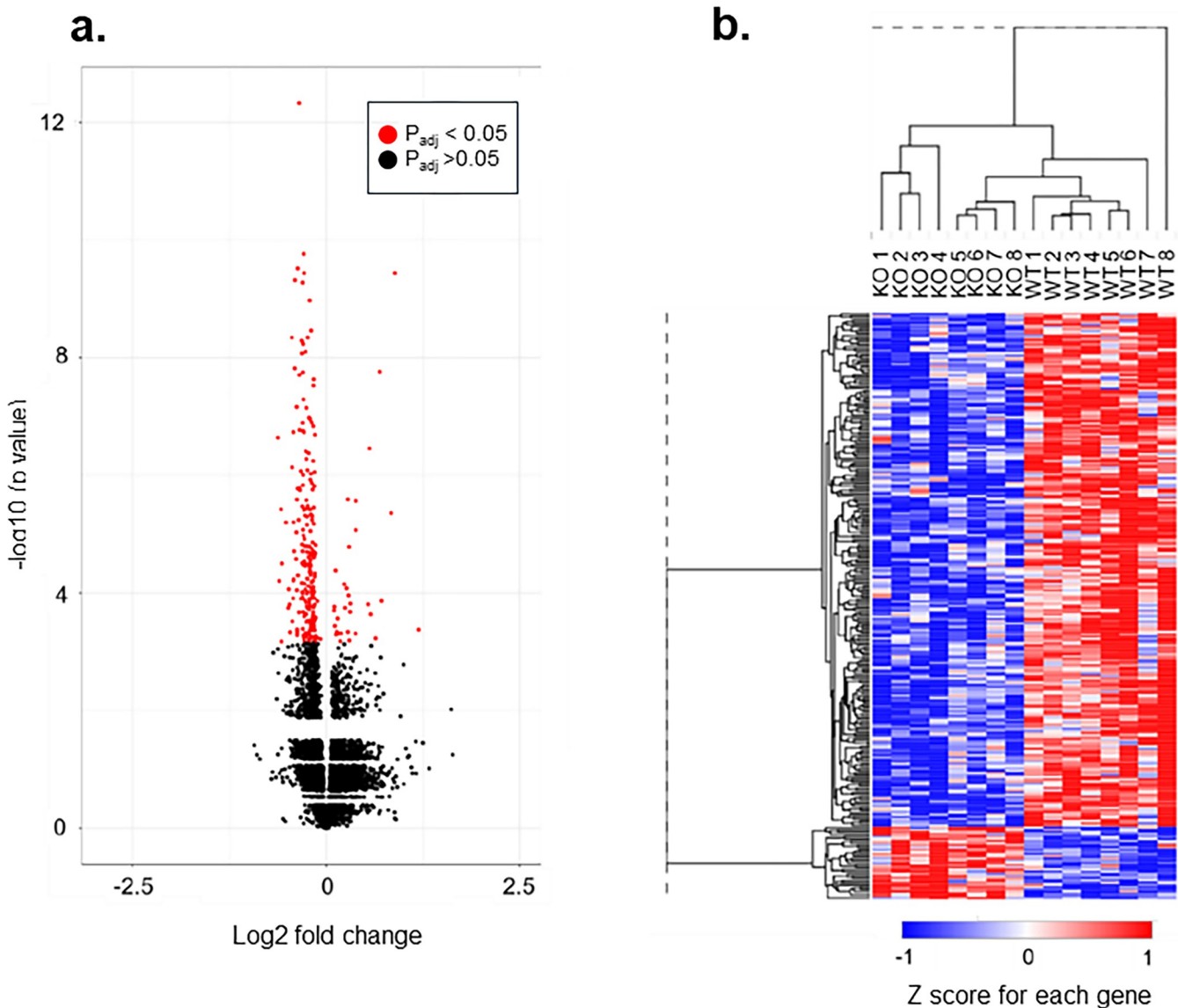

**Fig 2. Effects of *Setd1a* haploinsufficiency on transcriptional changes in E13.5 *Setd1a*$^{+/-}$ brain.** Volcano plot (a) shows all the differentially expressed genes discovered. There were 267 genes that were differentially expressed (234 downregulated and 33 upregulated), with log2 fold changes ranging from -0.63 to 1.19. A dendrogram showing hierarchical clustering (b) indicated that samples of the same genotype clustered together, suggesting similar patterns of gene expression changes within each group.

GO:0005739 (Benjamini Hochberg $p_{adj}$ = 0.002) survived correction for multiple testing. Within the initial analysis, however, were annotations relating to other mitochondrial terms, cilium, and methylation. Gene-set enrichment analysis revealed that genes that were differentially expressed in *Setd1a*$^{+/-}$ E13.5 brain were not significantly enriched for schizophrenia common variant association ($B$ = 0.004, SE = 0.088, p = 0.48).

### *Setd1a*$^{+/-}$ mice show an elevated acoustic startle response (ASR) and diminished prepulse inhibition (PPI)

Both male and female *Setd1a*$^{+/-}$ mice of Cohort 1 showed increased levels of startle responding to the 120 dB pulse-alone stimuli relative to WT littermates (Fig 3A, main effect of

**Table 1. Results of gene ontology (GO) term enrichment analysis.**

| Category | GO term | Gene count | *P value* | Benjamini Hochberg |
|---|---|---|---|---|
| Cellular component | Mitochondrion (GO: 0005739) | 47 | 0.00001 | 0.002 |
| | Cilium (GO: 005929) | 10 | 0.004 | 0.35 |
| | Ciliary basal body (GO: 0036064) | 7 | 0.005 | 0.30 |
| | Mitochondrial matrix (GO: 0005759) | 8 | 0.02 | 0.74 |
| | Integral component of mitochondrial inner membrane (GO: 0031305) | 3 | 0.04 | 0.80 |
| Biological process | Peptidyl-lysine deacetylation (GO: 0034983) | 3 | 0.006 | 0.98 |
| | Methylation (GO: 0032259) | 8 | 0.01 | 0.99 |
| | Cilium morphogenesis (GO: 0060271) | 8 | 0.01 | 0.95 |
| | Cell projection organization (GO: 0030030) | 7 | 0.02 | 0.97 |
| | Cilium assembly (GO: 0060271) | 6 | 0.04 | 1.0 |
| | Nucleotide metabolic process (GO: 0009117) | 3 | 0.04 | 1.0 |
| Molecular Function | Methyltransferase activity (GO: 0008168) | 8 | 0.009 | 0.95 |
| | Nucleic acid binding (GO: 0003676) | 26 | 0.01 | 0.80 |
| | NAD-dependent histone deacetylase activity (H3-K14 specific) (GO: 0032041) | 3 | 0.02 | 0.82 |
| | Transferase activity (GO: 0016740) | 29 | 0.02 | 0.84 |
| | Nucleotidyltransferase activity (GO: 0016779) | 5 | 0.03 | 0.86 |
| | Aspartate-tRNA ligase activity (GO: 0004815) | 2 | 0.03 | 0.82 |

GENOTYPE, $F_{1,73} = 5.35$, p = 0.02). Although there were no overall effects of SEX ($F_{1,73} = 3.25$, p = 0.08) or a GENOTYPE*SEX interaction ($F_{1,73} = 1.98$, p = 0.16) on ASR, the difference was much more pronounced in male mice ($t_{37} = 2.54$, p = 0.02 for comparison between male WT and $Setd1a^{+/-}$ mice), than female mice ($t_{36} = 0.67$, p = 0.51 for female comparison) suggesting greater impact of $Setd1a$ haplosufficiency on startle responsivity in male mice. This increase in ASR was relatively consistent throughout the session (S6 Fig). We were able to replicate the pattern of increased ASR in male mice, and also demonstrated a similar, but reduced increase in female mice in a second cohort of mice (S3A Fig), further validating increased startle responding as a result of $Setd1a$ haplosufficiency.

As expected, a weak prepulse stimulus was able to modify the ASR and increased prepulse amplitude led to greater PPI shown by the mice (Fig 3B, main effect of PPI-INTENSITY, $F_{1,73} = 253.7$, p<0.001), however $Setd1a^{+/-}$ mice showed reduced levels of PPI relative to WT (main effect of GENOTYPE, $F_{1,73} = 26.64$, p<0.001). This effect was consisted at both prepulse intensities used (GENOTYPE*PPI-INTENSITY interaction ($F_{1,73} = 2.00$, p = 0.16) and in both male and female $Setd1a^{+/-}$ mice (GENOTYPE*PPI-INTENSITY*SEX interaction, $F_{1,73} = 0.001$, p = 0.97). Comparisons between the groups of mice, demonstrated larger effects between female WT and $Setd1a^{+/-}$ mice ($t_{36} = 3.82$ and 4.75, p<0.001, for 8 and 16db prepulse stimuli, respectively) compared to male mice ($t_{37} = 2.04$, p = 0.049 and $t_{37} = 2.88$, p = 0.07 for 8 and 16db prepulse stimuli, respectively) suggesting greater impact of $Setd1a$ haplosufficiency on PPI in female mice. This pattern of attenuated PPI in both sexes of $Setd1a^{+/-}$ mice, was also replicated in Cohort 2 (S3B Fig). Thus, these data indicate that $Setd1a$ haplosufficiency led to robust phenotypes of hyperstartling and reduced PPI, with stronger effects on startle in male mice and PPI in female mice.

## Male $Setd1a^{+/-}$ mice are insentitive to the effects of risperidone on ASR, but disrupted sensorimotor gating in $Setd1a^{+/-}$ mice cannot be rescued by either risperidone or haloperidol

To investigate the effects of $Setd1a$ haplosufficiency on ASR and PPI further, and also to further explore the possible links between $Setd1a$ and schizophrenia, we used the antipsychotic

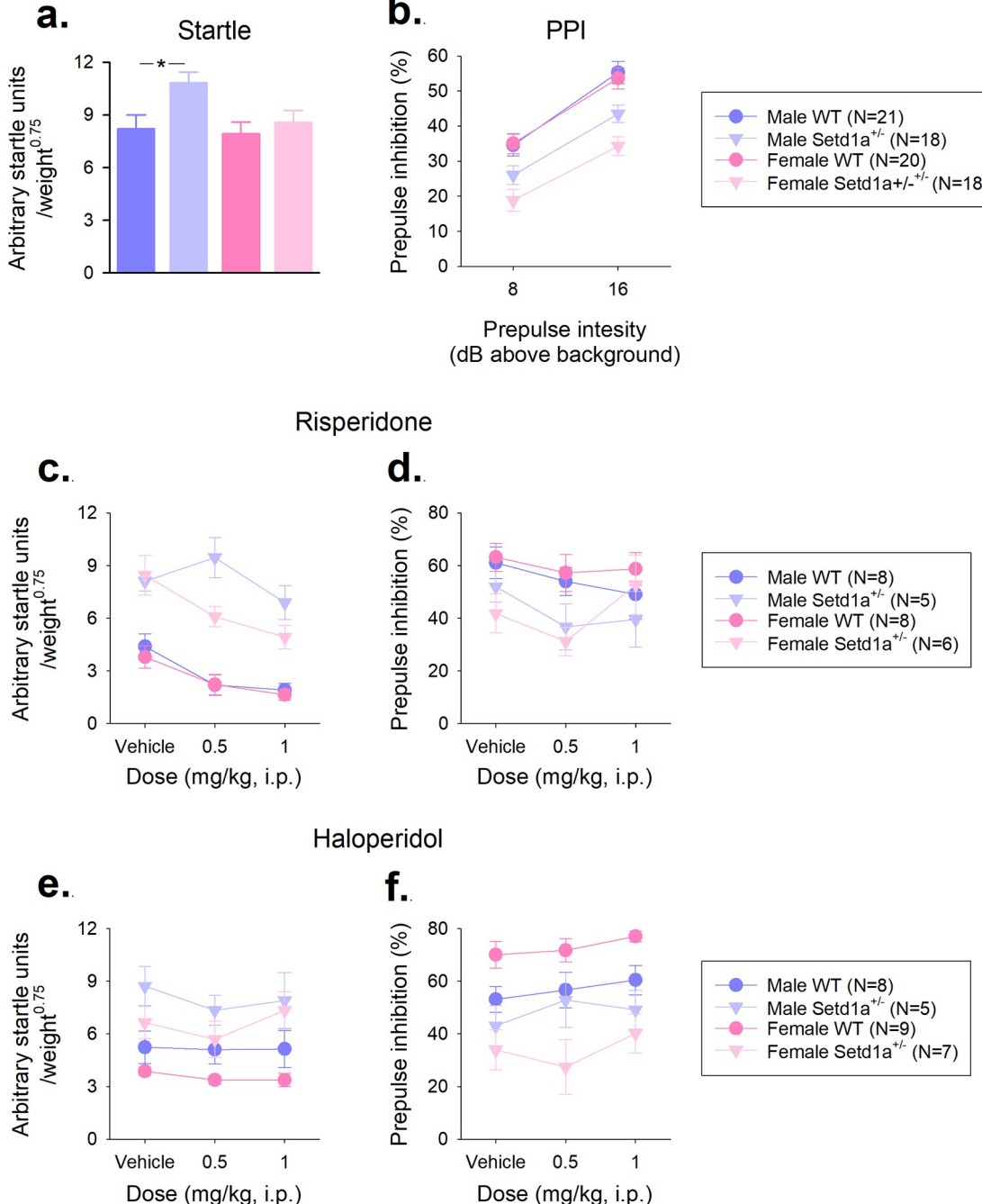

**Fig 3. *Setd1a*^+/-^ mice show an elevated acoustic startle response (ASR) and diminished prepulse inhibition (PPI), which cannot be rescued by haloperidol or risperidone.** Using the first cohort of mice, (a) mean responding to 120 dB pulse-alone startle stimuli (see S6 Fig for individual trial data) and PPI (b), the modification of the ASR when the startle stimuli were preceded by a prepulse at either 8 or 16 dB above background level (70dB) were assessed. In a separate naive group of mice (Cohort 2), we confirmed these patterns of elevated startle and reduced PPI (S3 Fig), before assessing the effects of the common antipsychotic drugs risperidone (c and d) and haloperidol (e and f) on ASR and PPI. Notes: only the data from 120 dB pulse alone and the 8 and 16 dB prepulse trials are reported here for brevity. For the drug studies, responses in the 8 and 16 dB prepulse trials were combined to generate a single value for ease of analysis and Cohort 2 was divided into separate groups for each drug study. * shows significant main effect of GENOTYPE at p<0.05. Data shows mean±SEM.

drugs risperidone or haloperidol with a second naïve cohort of mice (Cohort 2), divided into two groups; one for each drug. Following a series of pilot studies (S4 Fig) we used a dose of each drug (0.5mg/kg) which we had demonstrated to be sub-threshold in WT mice for both ASR and PPI. We compared the effects at 0.5mg/kg for each drug to vehicle, and also a higher dose (1mg/kg), where we showed risperidone decreased ASR and haloperidol increased PPI in WT mice in pilot work.

A significant significant three way interaction between GENOTYPE, DOSE and SEX ($F_{2,46}$ = 5.51, p = 0.01, $\mu^2$ = 0.19), suggested complex effects of risperidone on the startle response (Fig 3C). Investigating this further in males, the main effect of DOSE ($F_{2,22}$ = 7.32, p = 0.004, $\mu^2$ = 0.40) was qualified by a significant interaction between DOSE and GENOTYPE ($F_{2,22}$ = 6.26, p = 0.01, $\mu^2$ = 0.36). Bonferroni-corrected pairwise comparisons revealed a dose-dependent reduction of ASR in WT (relative to vehicle, p = 0.05 at 0.5 mg/kg and p = 0.01 at 1.0 mg/kg) but not *Setd1a*$^{+/-}$ males (relative to vehicle, *p* = 0.62 at 0.5 mg/kg and *p* = 0.42 at 1.0 mg/kg). Conversely, in females the main effect of DOSE was significant ($F_{2,24}$ = 23.09, p<0.001, $\mu^2$ = 0.66), but there was no GENOTYPE*DOSE interaction ($F_{2,24}$ = 1.32, p = 0.29). These findings suggest that the startle-inhibiting effect of risperidone, was blunted in male (but not female) *Setd1a*$^{+/-}$ mice relative to WT, a result confirmed by investigation of the relative change from vehicle with each dose of the drug (S7 Fig). Risperidone did not have any significant effects on PPI (Fig 3D, main effect of DOSE, $F_{2,46}$ = 2.71, p = 0.08), and there were no interactions between the dose of the drug and *Setd1a* haplosufficiency ($F_{2,46}$ = 1.38, p = 0.26), however the previously observed reduction of PPI in *Setd1a*$^{+/-}$ mice was maintained (main effect of GENOTYPE ($F_{1,23}$ = 6.36, p = 0.02, $\mu^2$ = 0.22). Therefore, these results demonstrate that risperidone did not rescue the reduction in PPI resulting from *Setd1a* haplosufficiency in either sex, however, the elevated startle response in *Setd1a*$^{+/-}$ male mice was insensitive to the modifying effects of the drug.

The elevated startle response shown by *Setd1a*$^{+/-}$ mice was unaffected by haloperidol (main effect of GENOTYPE, $F_{2,50}$ = 14.01, p = 0.001, $\mu^2$ = 0.40) at all doses used (GENOTYPE*DOSE interaction: $F_{2,50}$ = 1.72, p = 0.19) and with both sexes of mice (GENOTYPE*SEX interaction $F_{2,50}$ = 0.02, p = 0.90). Haloperidol did not significantly alter the startle response from vehicle (Fig 3E, main effect of DOSE, $F_{2,50}$ = 3.17, p = 0.05). Haloperidol was also without significant effect on PPI (Fig 3F, main effect of DOSE, $F_{2,50}$ = 3.08, p = 0.06), and the significant reduction of PPI in *Setd1a*$^{+/-}$ mice (main effect of GENOTYPE, $F_{1,25}$ = 21.92, p<0.001, $\mu^2$ = 0.47) was maintained across all doses (GENOTYPE*DOSE interaction, $F_{2,50}$ = 0.02, p = 0.98). There was no effect of SEX on these data ($F_{1,25}$ = 0.03, p = 0.87). Thus, these results show that neither *Setd1a* haplosufficiency induced elevated startle responses or attenuations of PPI were normalised by haloperidol, in either sex, confirmed by analysis of the relative change from vehicle with each dose of the drug (S7 Fig).

### Male and female *Setd1a*$^{+/-}$ mice show increased anxiety-related behaviour

In the 10 minutes open field test (OFT), all mice (from Cohort 1), as expected, tended to avoid the inner zone of the arena (Fig 4A, main effect of ZONE, $F_{1,72}$ = 5736.72, p<0.001), however a significant GENOTYPE*ZONE interaction ($F_{1,72}$ = 4.58, p = 0.04) indicated that *Setd1a*$^{+/-}$ mice explored the central zone less (Fig 4B, main effect of GENOTYPE, $F_{1,72}$ = 4.58, p = 0.036). Female mice showed increased avoidance of the central zone (main effect of SEX, $F_{1,72}$ = 5.69, p = 0.02), a pattern of behaviour that was consistent in both groups of mice (GENOTYPE*SEX interaction, $F_{1,72}$ = 0.15, p = 0.70). This demonstrates that both sexes of *Setd1a*$^{+/-}$ mice had a reduced tendency to explore the more anxiogenic central region of the arena. This was supported by reduced transitions into the central zone (Fig 4C, main effect of

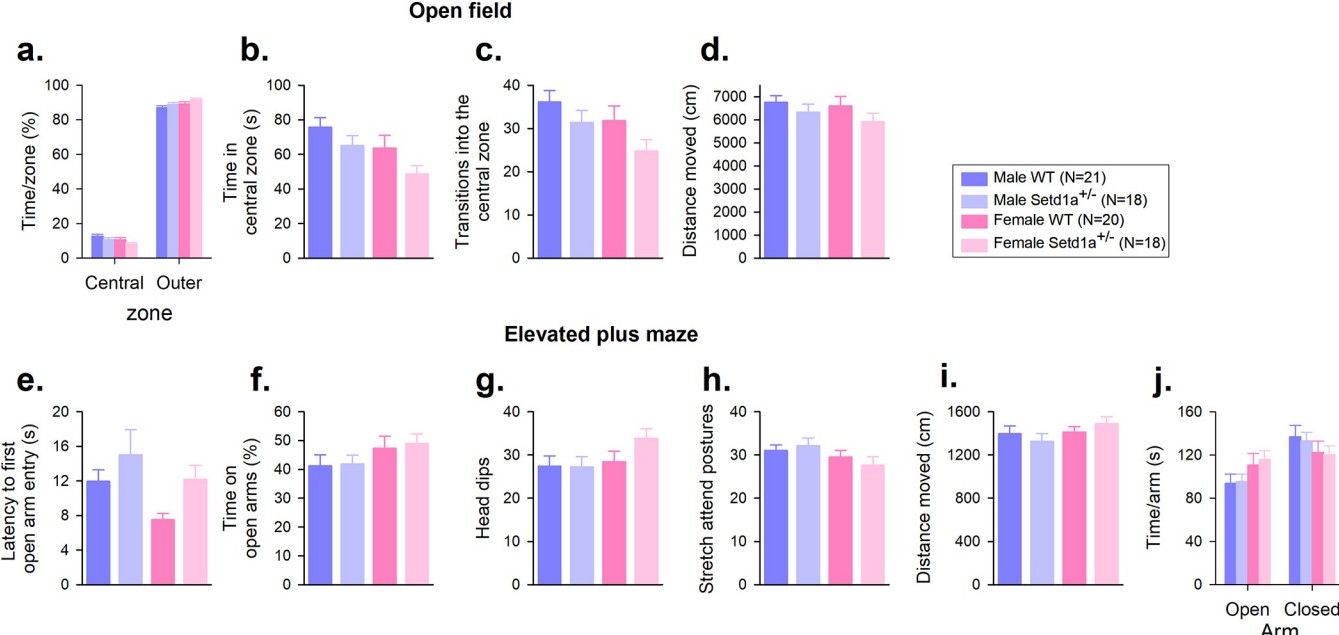

**Fig 4. *Setd1a*<sup>+/-</sup> mice show increased anxiety-related behaviour in the open field test (OFT), but not the elevated plus maze (EPM).** All mice showed an anxiogenic profile in the OFT with reduced exploration of the anxiety-inducing central zone (a). *Setd1a*<sup>+/-</sup> showed reduced time (b) and entries (c) into this region. There were no differences in the overall distance moved throughout the arena (d). Data from one female WT animal were excluded from analysis of the OFT because it was identified as a multivariate outlier. *Setd1a*<sup>+/-</sup> mice also showed an anxiogenic profile on EPM, with a slower latency to enter the more anxiety-inducing open arms (e). Other measures: time on open arms (f), head dips (g), stretch attend postures (h) and the overall distance moved on the maze (i) demonstrate an anxiogenic profile on the EPM by all groups of mice. Data shows mean±SEM. All data from Cohort 1 mice.

GENOTYPE, $F_{1,72}$ = 4.08, p = 0.047). There were no differences between *Setd1a*<sup>+/-</sup> and WT mice in the total distance moved (locomotion) during the OFT (Fig 4D, main effect of GENOTYPE, $F_{1,72}$ = 2.40, p = 0.13) suggesting an equivalent propensity to explore the arena, even though *Setd1a* haplosufficiency in both males and females led to more anxiogenic behaviour.

*Setd1a*<sup>+/-</sup> mice from Cohort 1 also showed an anxiogenic profile on the elevated plus maze (EPM), with a slower latency to enter the more anxiety-inducing open arms of the EPM (Fig 4E, main effect of GENOTYPE, $F_{1,73}$ = 4.75, p = 0.032) although over the entire 5 min session they ultimately explored the open arms an equivalent amount to their WT littermates (Fig 4F, main effect of GENOTYPE, $F_{1,73}$ = 0.09, p = 0.76), and demonstrated equal amounts of anxiety-induced behaviour such as head dips (Fig 4G, main effect of GENOTYPE, $F_{1,73}$ = 1.20, p = 0.28), and stretch attend postures (Fig 4H, main effect of GENOTYPE, $F_{1,73}$ = 0.05, p = 0.82). There were no differences between *Setd1a*<sup>+/-</sup> and WT mice in the total distance moved during the test (Fig 4I, main effect of GENOTYPE, $F_{1,73}$ = 0.002, p = 0.97) suggesting an equivalent propensity to explore the maze, and as expected, all of the mice showed an avoidance of the more anxiety-inducing open arms of the EPM (Fig 4J, main effect of ARM, $F_{1,73}$ = 7.41, p = 0.008), and there were no significant effects of SEX or interactions.

## Locomotor activity levels, motoric function, and novel object recognition are not altered in *Setd1a*<sup>+/-</sup> mice

Using Cohort 1, locomotor activity was assessed over three consecutive days, with a 2 hour session each day. The number of beam breaks made decreased each day demonstrating habituation (Fig 5A, main effect of Day, $F_{1.69,123.08}$ = 28.63, p<0.001) and no significant

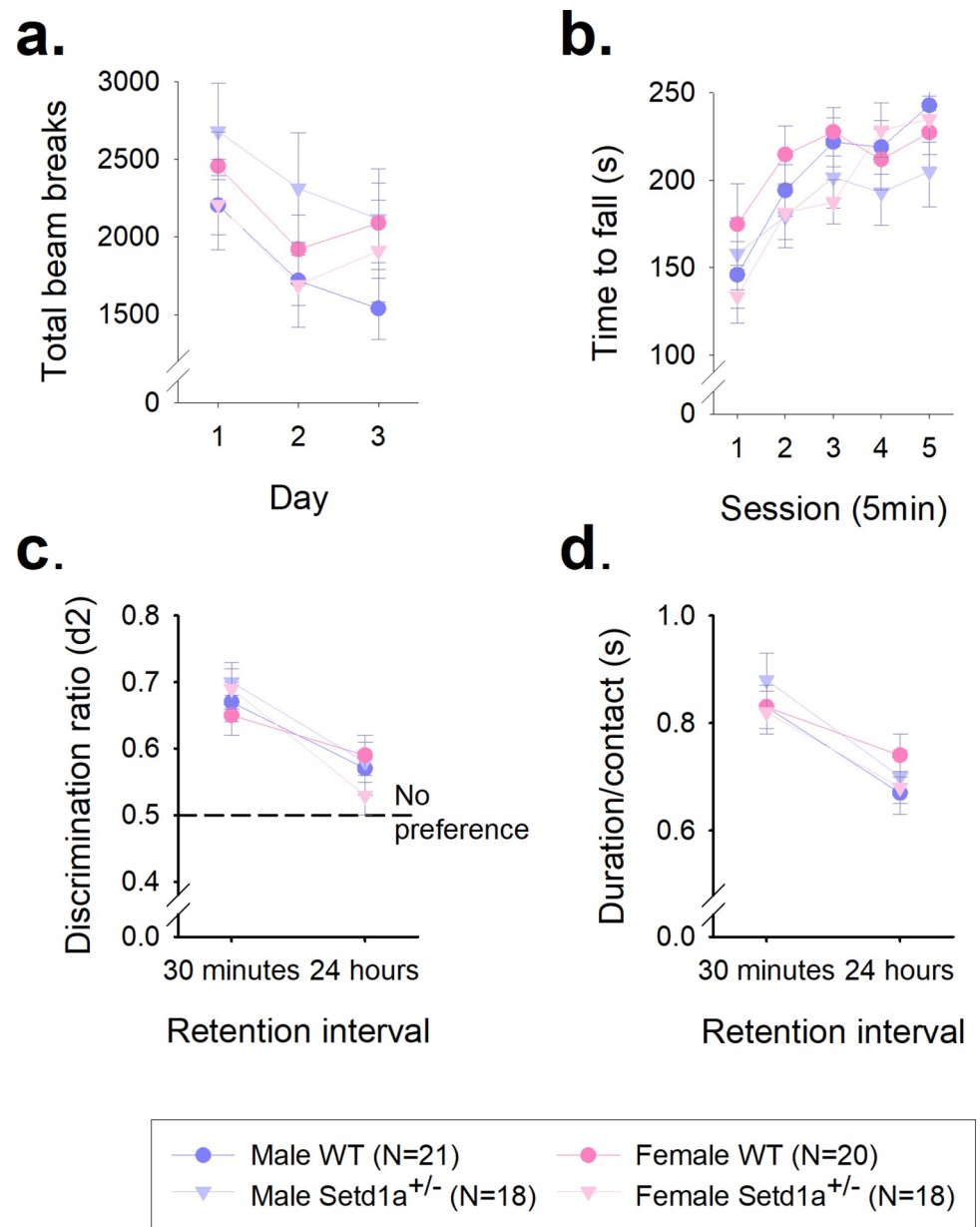

**Fig 5. *Setd1a*$^{+/-}$ mice show normal locomotor activity levels, motoric function, and novel object recognition (NOR) memory.** Locomotor activity measured over 3 days (a) and assessment of motor co-ordination and learning on the rotorod (b). Discrimination raito (c) and object contact times (d) from the NOR test sessions, after 30 min and 24 hours delay. Data from one male WT subject was missing from the NOR due to a technical issue during testing. Data shows mean±SEM.

GENOTYPE*DAY interaction ($F_{1.69,123.08} = 0.18$, $p = 0.80$), showed that the degree of habituation of locomotor activity levels across days was equivalent in *Setd1a*$^{+/-}$ and WT mice. *Setd1a* haploinsufficiency did not alter the overall amount of activty over the three sessions (main effect of GENOTYPE, $F_{1,73} = 0.46$, $p = 0.50$). There was no significant main effect of SEX ($F_{1,73} = 0.40$, $p = 0.84$), and no significant interactions between GENOTYPE, SEX and DAY (all $p>0.05$). These data would suggest that both male and female *Setd1a*$^{+/-}$ mice show normal locomotor activity levels and habituation of activity levels across test sessions.

Assessment of motor co-ordination using the Rotarod revealed that the latency to fall from the rotating beam over five accelerating trials was not significantly different between $Setd1a^{+/-}$ and WT mice (Fig 5B, main effect of GENOTYPE, $F_{1,73} = 2.70$, $p = 0.11$). A significant main effect of TRIAL ($F_{2.97,217.0} = 16.99$, $p<0.001$), indicated that performance increased with training, which was equivalent between $Setd1a^{+/-}$ and WT mice (GENOTYPE*TRIAL interaction $F_{2.97,217.0} = 0.50$, $p = 0.68$). There was also no significant main effect of SEX ($F_{1,73} = 0.33$, $p = 0.57$) and no significant GENOTYPE*SEX interaction ($F_{1,73} = 0.001$, $p = 0.98$). The GENOTYPE*SEX*TRIAL interaction was also not significant ($F_{2.97,217.0} = 2.41$, $p = 0.07$). These data indicate normal motor co-ordination and motor learning in male and female $Setd1a^{+/-}$ mice.

Novel object recognition memory was assessed after a 30 minute and 24 hour retention interval, and as expected, discrimation ratios were higher after 30 minutes compared to 24 hours (Fig 5C, main effect of DELAY, $F_{1,72} = 19.61$, $p<0.001$). There was no significant main effect of GENOTYPE ($F_{1,72} = 0.05$, $p = 0.82$) and no GENOTYPE*DELAY interaction ($F_{1,72} = 1.68$, $p = 0.20$), suggesting no differences in recall by $Setd1a^{+/-}$ mice. There were no significant effects of SEX ($F_{1,72} = 0.28$, $p = 0.61$) or significant interactions between these different factors on the discrimation ratios. Object exploration in these sessions, indexed by duration per contact (Fig 5D) also showed no differences between WT and $Setd1a^{+/-}$ mice (main effect of GENOTYPE, $F_{1,72} = 0.01$, $p = 0.97$), although all mice showed a reduced tendency to explore the objects in the 24 hours test (main effect of DELAY, $F_{1,72} = 27.13$, $p<0.001$). There were also no significant effects of SEX ($F_{1,72} = 0.01$, $p = 0.97$) or interactions between the different factors of the ANOVA. Analysis of the acquisition time data (S3 Table) revealed no significant effect of GENOTYPE on the time taken to achieve 40 seconds of object exploration ($F_{1,72} = 0.09$, $p = 0.76$) or between the two sessions ($F_{1,72} = 0.12$, $p = 0.091$), suggesting that all the mice explored the objects equally in both delay conditions. These findings indicate intact short- and long-term object recognition memory in both male and female $Setd1a^{+/-}$ mice.

## Discussion

Here, we examined the developmental and behavioural consequences of haploinsufficiency of *Setd1a* in a novel mouse model. We provide evidence for a complex pattern of sex-related differences spanning the pre- and post-natal period: reduced placental size in males and females at different foetal ages, post-natal catch-up growth in males, and effects on adult behaviour. Whole brain RNA-Seq analysis at E13.5 substantiate previous findings on the impact of Setd1a loss of function (LoF) on markers of mitochondrial function. With a literature demonstrating *Setd1a* as a risk candidate for a schizophrenia, we observed a number of phenotypes of relevance, including increased anxiety-related behaviour, enhanced acoustic startle response, and decreased pre-pulse inhibition of acoustic startle. These abnormalities strengthen the support for the use of *Setd1a* haploinsufficient mice as a model for the biological basis of schizophrenia, although the neural mechanisms underpinning this may be complex and involve pre-natal effects.

Using the same model of *Setd1a* LoF as we have used here, Clifton et al., 2022 [25] showed gene expression patterns in cortical samples from E14 through to P70 with consistent downregulation in genes enriched for mitochondrial function. We have replicated these findings at a similar developmental time point and demonstrate a similar relationship with mitochondrial GO terms. Altered mitochondrial gene expression changes have also been observed in an RNAi SETD1A knockdown cellular model [4, 35] and mitochondrial dysfunction has been repeatedly implicated in the pathogenesis of schizophrenia [36–39]. A recent study using heterozygous disruption of SETD1A in hiPSC-derived neurones demonstrated a metabolic

dysfunction that affected neurite growth and differentiation [40]. Common variants for schizophrenia are dynamically expressed during neurodevelopment, with high expression in foetal brain [41–43], and the cellular heterogeneity of the E13.5 mouse brain may have masked subtle gene expression changes, and also may have reduced the number of differentially expressed genes we were able to observe, in comparison with previous studies [17, 19, 25]. However, we were also able to identify differential expression in several nominally significant cilium-related GO terms, in addition to the mitochondrial markers. Cilia play an essential role during neurodevelopment [44] and knockdown of a range of neuropsychiatric risk genes affects cilia formation *in vitro* [45]. Taken together, these findings provide further evidence that *Setd1a* haploinsufficiency, early in development, may have a more generic effect on biological mechanisms, rather than specifically targeting pathways known to be implicated in schizophrenia.

Our data demonstrate that foetal growth was unaffected by *Setd1a* haploinsufficiency through gestation, with the expected weight increases up to birth. However, there were some interesting differences in placental weight with, on average, smaller placentae in *Setd1a*$^{+/-}$ embryos across all time points evaluated, which led to increased embryo:placenta ratios (EPR) in *Setd1a*$^{+/-}$ embryos relative to WT. These differences were most notable at E11.5, where male *Setd1a*$^{+/-}$ placentae were ~32% smaller than WT and at E18.5 where female *Setd1a*$^{+/-}$ placentae were ~16% smaller than both male and female WT placentae. Smaller sizes may reflect impairment in function, and with the placenta acting as the vital conduit for nutrition, this could imply a paucity of essential substances, such as neurotransmitters and amino acids, at key developmental stages [46]. This may be further impacted by the fact that foetal size remains unaltered, which could suggest an imbalance between foetal demands and the ability of the placentae to match this need, which could inhibit developmental progression [47]. For example, using a mouse model of *Igf2* dysfunction, where only placental expression was reduced [48], we demonstrated that a foetus:placenta imbalance can lead to adult-related changes in anxiety-related behaviours [47]. In this model, placental size is reduced (~25% E12 to E19) but the foetus only begins to show intrauterine growth restriction (IUGR) late on at E19 leading to a low birth weight (31% of WT, a period of catch-up growth to match adult weight by P100 [47, 48]. Other models show similar patterns [46]. Thus, a mismatch between foetal demands and placental function can lead to IUGR, low birth weight, a period of exaggerated early (catch-up) growth and adult-related changes in behaviour. We observed similar patterns of pre-natal and post-natal growth indices in our model of *Setd1a* haploinsufficiency, although we did not measure birth and early post-natal weights, but male mice demonstrated a period of catch-up growth adult. We also observed more complex effects suggesting a sex-related change in placental function, which could affect later behaviour in a sex-related manner [46, 49]. IUGR, as an indicator of placental dysfunction has been implicated in a number of neuropsychiatric conditions, such as schizophrenia, autism and other neurodevelopmental disorders, with the association of conceptus sex also influencing placental (dys)functionality [46, 50–52].

In terms of effects on adult behaviour, we demonstrated comparable patterns on acoustic startle responding (ASR) and the gating of this response by a prepulse stimulus (prepulse inhibition, PPI), to previous studies in *Setd1a* haploinsufficiency mice [18]. Deficits in attention and gating, or filtering out intrusive stimuli, are prominent features shared by both psychiatric and developmental disorders [53–55], with which haploinsufficiency of *SETD1A* is associated [14, 22], therefore the consistent effect from our and other's work [17, 18] in *Setd1a* haploinsufficiency mice of disrupted PPI may be expected. Altered ASR in male *Setd1a*$^{+/-}$ mice only was also found by Chen et al [18]. No effects on ASR were found in the Naga et al study [17], but the sex of the mice is not clear from the methods, and results are not split by sex. We

replicated our initial finding in a separate cohort of male and female mice (Supplementary Methods Fig 2) thus, it is possible to infer a sex-dependent effect on increased ASR and a sex-independent effect on PPI. In terms of startle responding, the elevated response we observe in male $Setd1a^{+/-}$ mice may reflect initial sensitisation to a fearful stimulus, and reduced habituation to the startle stimuli through the test session. Increased startle responding may be related to an altered fear response and/or altered emotional processing via involvement of the amygdala [56] and may reflect hypervigilance [57] which could link to the subtle anxiogenic effects we observed. There is some evidence for an interaction between sex and early life adversity, as demonstrated by the $Igf2$-P0 IUGR model previously discussed [47], and with male rats demonstrating increased ASR following maternal separation [58, 59] that could also be relevant to the developmental changes we have observed. Increased startle responding has been observed in neurodevelopmental conditions such as autism and Fragile X syndrome [53], in addition to PTSD [60] and schizophrenia [61]. These conditions commonly present more in males than females, so this phenotype might be expected, however, studies have more commonly observed increased startle responding in female subjects compared to males [58, 62].

Furthermore, we also demonstrated sex-dependent sensitivity to the typical anti-psychotic drug risperidone, with male $Setd1a^{+/-}$ mice showing a blunted effect to the drug on the ASR, whereas the drug reduced ASR in a dose-dependent way in WTs and female $Setd1a^{+/-}$ mice. However, there is little evidence for sex-related effects of acute risperidone administration, but chronic dosing appears to show a female bias, with female rats appearing to be more sensitive to chronic dosing at a young age [63] and female schizophrenia patients responded better to chronic risperidone treatment than male patients [64]. The mode of action of risperidone, with high activity via serotonergic 5-HT$_{2A}$ receptors [64] suggests an interesting target to investigate in the future, especially in male $Setd1a^{+/-}$ mice. Haloperidol, also a typical anti-psychotic drug, though mainly dopaminergic in function [65] had no effects on the ASR in male (or female) $Setd1a^{+/-}$ mice. Sensorimotor gating deficits (PPI) could not be rescued by haloperidol or risperidone, although both anti-psychotics have previously been shown to reverse PPI deficits in several rodent models of schizophrenia [33]. This suggests that both these anti-psychotic agents are ineffective for ameliorating schizophrenia-relevant phenotypes in $Setd1a^{+/-}$ mice, of both sexes, and point to deficits in neural systems other than the monoamine system which may be consistent with variable responses to antipsychotic treatment in individuals with $SETD1A$ variant-associated schizophrenia [16]. Of particular relevance here, is the similarity of the disrupted PPI shown by $Setd1a^{+/-}$ LoF models and the effects of NMDAR antagonists, such as PCP, MK-801 and ketamine [66], which taken together with the possibility that glutamate release is likely reduced when $Setd1a$ is knocked down in postsynaptic pyramidal neurons [17], suggests that the glutamatergic system may be compromised as a result of Setd1a haploinsufficiency [21].

In terms of measures of anxiety, then our data from the open field (OFT) and elevated plus maze (EPM) assays are the first to demonstrate that $Setd1a$ LoF can lead to a subtle anxiogenic phenotype. Exploration patterns in these tests demonstrated that both male and female $Setd1a^{+/-}$ mice showed avoidance of the more anxiety-inducing regions of each piece of apparatus, the central zone of the OFT and the open arms of the EPM. Previous studies using the OFT [17–19] have used a procedure more akin to our locomotor activity assessment, with repeated long sessions and found, as we have, that basic motoric behaviour and investigation is not affected. Within session measures of activity from the OFT and EPM also confirm this. It would be interesting if in these studies [17–19] the first few minutes of the first sessions could be re-examined to gauge if any anxiety phenotype was present. Taken together, these data illustrate subtle effects of $Setd1a$ LoF on anxiety, which is consistent with emotional issues as co-morbid presentations in schizophrenia [67]. The other adult behaviour we investigated

recapitulated previous findings in *Setd1a*$^{+/-}$ haploinsufficient mice, with no significant effects on novel object discrimination and recall (NOR), and no profound motoric impairments in either male or female mice. NOR performance was equivalent to Mukai et a [19] with no differential recall with a short or long retention delay. However, in the Nagahama et al [17] study there was an impairment at the short delay, but not the longer timepoint possibly indicating a subtle phenotype that is subsumed by the difference between their model, and the ones used here and by Mukai et al [19] which may reflect the different targeting strategies, and loci, for disrupting *Setd1a* [21].

Taken together these data demonstrate that this new *Setd1a* LoF mouse model recapitulates some of the core findings of previous models, of relevance to neurodevelopmental disorders and schizophrenia in a large experimental cohort of both sexes. We also provide some insight into *Setd1a* haploinsufficiency on development and how it might interact with sex. The neurodevelopmental hypothesis of schizophrenia highlights the pre-and peri-natal phase of life as key period in the onset of the disorder, with intrauterine compromise an important factor [68, 69]. Sex differences in schizophrenia are well documented, in terms of incidence and symptom presentation [70], thus the altered developmental indices in *Setd1a*$^{+/-}$ conceptuses and behavioural differences we found add a further dimension to this theory. Taken together our findings would suggest pre-natal effects of *Setd1a* LoF on adult behaviour, mediated by impaired placental functionality and metabolic function in developing neurones. Use of a conditional KO model, where setd1a LoF can be "switched off" either during development or in adulthood would be one way to unpick this conundrum. However, either way, these data add to the growing understanding of the role of *Setd1a* in brain development and behaviour. Our findings, coupled with those from others investigating different *Setd1a* models, demonstrate replicable common phenotypes of relevance that could be used for the development and testing of therapies in rescue studies.

## Supporting information

**S1 Fig. Confirmation of *Setd1a* haploinsufficiency in the *Setd1a*$^{+/-}$ model.**
(DOCX)

**S2 Fig. Methods for RNAseq.**
(DOCX)

**S3 Fig. Replication of acoustic startle and prepulse inhibition effects in a separate cohort of *Setd1a*$^{+/-}$ and WT mice.**
(DOCX)

**S4 Fig. Pilot studies for pharmacological investigations.**
(DOCX)

**S5 Fig. Trajectory of Setd1a expression across neurodevelopment in WT mice and confirmation of *Setd1a* haploinsufficiency in the *Setd1a*$^{+/-}$ model.**
(DOCX)

**S6 Fig. Acoustic startle response analysis per trial.**
(DOCX)

**S7 Fig. Change from vehicle following treatment with risperidone or haloperidol.**
(DOCX)

**S1 Table. Samples sizes for evaluation of embryo and placenta size.**
(DOCX)

**S2 Table. Details of the litters, genotype and sex ratios of the samples used in the pre- and postnatal assessments.**
(DOCX)

**S3 Table. Supporting data for the novel object recognition test.**
(DOCX)

## Acknowledgments

The MRC Mary Lyon Centre (Harwell, Oxfordshire, UK) generated the C57BL/6NTac-Setd1atm1c(EUCOMM)Wtsi/WtsiCnrm mouse strain from which we derived the strain used in this study.

## Author Contributions

**Conceptualization:** Matthew L. Bosworth, Anthony R. Isles, Lawrence S. Wilkinson, Trevor Humby.

**Data curation:** Matthew L. Bosworth, Anthony R. Isles, Trevor Humby.

**Formal analysis:** Matthew L. Bosworth, Trevor Humby.

**Funding acquisition:** Matthew L. Bosworth, Lawrence S. Wilkinson, Trevor Humby.

**Investigation:** Matthew L. Bosworth, Anthony R. Isles, Trevor Humby.

**Methodology:** Matthew L. Bosworth, Anthony R. Isles, Trevor Humby.

**Project administration:** Matthew L. Bosworth, Anthony R. Isles, Lawrence S. Wilkinson, Trevor Humby.

**Supervision:** Anthony R. Isles, Lawrence S. Wilkinson, Trevor Humby.

**Writing – original draft:** Matthew L. Bosworth, Anthony R. Isles, Trevor Humby.

**Writing – review & editing:** Matthew L. Bosworth, Anthony R. Isles, Lawrence S. Wilkinson, Trevor Humby.

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
