## [Decision Letter · Decision Letter 0]

28 May 2024

PONE-D-24-03923Sex-dependent effects of Setd1a haploinsufficiency on development and adult behaviourPLOS ONE

Dear Dr. Humby,

Thank you for submitting your manuscript to PLOS ONE. After careful consideration, we feel that it has merit but does not fully meet PLOS ONE’s publication criteria as it currently stands. Therefore, we invite you to submit a revised version of the manuscript that addresses the points raised during the review process.

We look forward to receiving your revised manuscript.

Kind regards,

Kenji Tanigaki, Ph.D., M.D.

Academic Editor

PLOS ONE

Journal Requirements:

 [This work was supported by a Wellcome Trust Integrative Neuroscience PhD grant (109084/Z/15/Z) and an A. Bruce Naylor Memorial Early Career Research Fellowship from The Waterloo Foundation awarded to MLB; UKRI Medical Research Council (MRC) IMPC: Pump Priming Award (MR/P026176/1) awarded to ARI, LSW and TH; ARI, TH and LSW are members of the MRC Centre for Neuropsychiatric Genetics and Genomics (MR/L010305/1). ].  

5. Please include a copy of Table 1 and 2 which you refer to in your text on page 7 and 18.

Reviewers' comments:

Reviewer's Responses to Questions

**Comments to the Author**

1. Is the manuscript technically sound, and do the data support the conclusions?

Reviewer #1: Yes

Reviewer #2: Yes

2. Has the statistical analysis been performed appropriately and rigorously? 

Reviewer #1: Yes

Reviewer #2: Yes

3. Have the authors made all data underlying the findings in their manuscript fully available?

Reviewer #1: Yes

Reviewer #2: Yes

4. Is the manuscript presented in an intelligible fashion and written in standard English?

Reviewer #1: Yes

Reviewer #2: Yes

5. Review Comments to the Author

Reviewer #1: The manuscript describes the phenotype of a novel mouse model with a haploinsufficiency in Setd1a. Pre- and postnatal weights, placenta size, embryonic whole-brain RNAseq, as well as startle and PPI, open field and elevated plus maze testing was performed. A subset of animals were treated with haloperidol and risperidone.

The manuscript is very well written and the experimental approaches were sound in principle. I have several concerns and suggestions:

Major:

- I did not find any figure legends in the manuscript, which made it difficult to interpret some of the figures - was this a technical glitch or are they missing?

- The sample size for the drug treatment is very low, not sure how valid this data is. Also, the vehicle data does not correspond to the data in Figs 3a/b, nor supplemental figs 3. Why is that?

- Please mark all significance with an asterisk in all figures.

- Headers over the figure panels would be great to understand what is shown (especially in absence of legends!)

- Please consider adding dots for individual data points to some graphs, e.g. Fig. 3a

- Consider normalizing to vehicle condition for each mouse for the drug effect (Figs. 3c-e)

minor:

line 234: acronyms like EPM and OFT need to be introduced

line 290ff: were the experimenter blinded to the genotype of the mouse?

line 365ff: If a difference is not significant, there is no difference, just noise (unless the study is underpowered!). Please abstain from discussing "non-significant differences"

Reviewer #2: Overall, it is a very interesting study, analyzing the effects of Sedt1a haploinsufficiency in mice during early development and in adulthood. The study is well designed and all experimental details and results are nicely displayed and support the drawn conclusions. I only have some minor points that might further improve the study:

1. Novel object test: Was the observer blind to the condition of the animals?

2. In line 425 you mention table 1, but unfortunately this table is missing.

3. Figure legends are missing. Please add this to the manuscript, as the figures are not self-explaining.

4. Figure 2: To me it is unfortunately not clear what I should take out of this figure, despite the fact that some genes are upregulated, while others are downregulated. Maybe you could indicate here important genes that are altered in their expression profile, or at least the bottom-line of the dendrograms.

5. Fig. 3: Maybe you could add the respective drugs (risperidone, haloperidol) to the figure labels.

6. PLOS authors have the option to publish the peer review history of their article (what does this mean?). If published, this will include your full peer review and any attached files.

Reviewer #1: No

Reviewer #2: No

---

## [Author Response · Author response to Decision Letter 0]

24 Jun 2024

Reply to Reviewers' comments:

Reviewer #1

We thank the reviewer for their supportive comments and for highlighting some points that will help to improve the overall manuscript quality.

Major points:

1. I did not find any figure legends in the manuscript, which made it difficult to interpret some of the figures - was this a technical glitch or are they missing?

Reply: many apologies for this oversight, they got missed off during the submission process. They have been added at the end of the manuscript (pages 33 to 36)

2. The sample size for the drug treatment is very low, not sure how valid this data is. Also, the vehicle data does not correspond to the data in Figs 3a/b, nor supplemental figs 3. Why is that?

Reply: For the first issue relating to data validity, then the observed power for the reported analyses were high (>0.6) with 0.828 for the GENOTYPE*DOSE*SEX effect for risperidone, main effects of GENOTYPE for PPI for risperidone and haloperidol were 0.676 and 0.994, respectively. The power for the GENOTYPE effect on startle in the haloperidol experiment was 0.949. Furthermore, effect sizes (µ2) for the significant effects (main and interactions) were in the 0.35 to 0.7 range, representing medium/small to medium/large effects, demonstrating that although there was a low N in some groups the pattern of effects demonstrated can be interpreted as reliable to the hypotheses under test. We have added the effect size (µ2) values for the significant effects found following risperidone and haloperidol treatment, to aid interpretation of the findings (lines 496-523).

For the second point, we understand that the unfortunate absence of the legend may have made interpretation of this figure more difficult than it should have done. We had provided a description in the legend to explain the data pathway, indicating that the data for figures a and b, was from the first cohort of mice, the replication in untreated mice (Supplementary Information, S7 Fig) and drug study (figures c-f) from cohort 2. Therefore, with different mice used, different values would be obtained, however, it should be noted that, qualitatively, the results are the same. For the drug studies, the 2nd group of mice was randomly divided into separate groups for the two drugs, which might also account for some of the apparent variation shown, from the initial test data from Cohort 1 (Fig. 3a) and pre-drug assessment in Cohort 2 (S7 Fig). Although, qualitatively, the genotype-related effects observed were present with vehicle treatment. To make this more clear for the readers, we have emphasised the different studies that the two cohorts of mice were used in throughout the results section and figures (e.g. line 201, 333-334, 461, 476-477) and the 2nd cohort was divided into two groups for the drug study (lines 333-334, 488, 531, 547, 563).

3. Please mark all significance with an asterisk in all figures.

Reply: Thank you for this suggestion, but we have already identified significant genotype differences in the figures in this way, indicating other significant differences (i.e. task effects) could confuse the clarity of the figures. 

4. Headers over the figure panels would be great to understand what is shown (especially in absence of legends!)

Reply: Thank you for this suggestion to aid clarity, we have therefore labelled each of the figures in the main report and the Supplementary Information.

5. Please consider adding dots for individual data points to some graphs, e.g. Fig. 3a

Reply: For some of the figures this would not be appropriate, make them too busy and decrease clarity, but we have added data points to Fig. 3a, as per your suggestion. 

6. Consider normalizing to vehicle condition for each mouse for the drug effect (Figs. 3c-e)

Reply: we chose not to present the data in this way as it had the potential to obscure the real values of the startle and PPI responses and mask the significant effects of elevated baseline startle responding we observed. However, we do acknowledge that calculating relative change to vehicle does provide additional information about the amount of difference the different drugs/doses are exerting, therefore we have added these data to the supporting information (S7 Fig). As expected, the pattern of effects did not differ substantially from the main analysis presented, such that risperidone led to significant differences in the effects of the drug on startle in the male Setd1a+/ mice only. Whereas both doses of the drug decreased the startle response in male and female WT and female Setd1a+/ mice, 0.5 mg/kg led to an increase in startle amplitude in male Setd1a+/ mice. Consistent with the main findings, this analysis further demonstrated that there were no effects of risperidone on PPI, or haloperidol on startle or PPI. We have presented these data as Supplementary Information (S7 Fig), and cited them from the main text, for example “, a result confirmed by investigation of the relative change from vehicle with each dose of the drug (S7 Fig).”. (lines 507 and 527-528).

minor:

1. line 234: acronyms like EPM and OFT need to be introduced.

Reply: Thank you for pointing this out, full names have been included at this point in the text (line 246) and other places in the manuscript and figures.

2. line 290ff: were the experimenter blinded to the genotype of the mouse?

Reply: Thank you for pointing this out, and yes, the experimenter was blinded to the mice genotypes during the different novel object test (and the other experiments too). We have added a comment to this point (lines 199-201, 294-295).

3. line 365ff: If a difference is not significant, there is no difference, just noise (unless the study is underpowered!). Please abstain from discussing "non-significant differences"

Reply: This point relates to the Supplementary Information table (S1 Table) which describes litter sizes and genotype and sex ratios. The aim here was to demonstrate that at each age point these data remained relatively consistent. We have clarified the writing here to remove the discussion of non-significant effects (lines 375-380).

Reviewer #2:

We thank the reviewer for their supportive comments and for highlighting some minor points that will help to improve the overall manuscript quality and clarity.

1. Novel object test: Was the observer blind to the condition of the animals?

Reply: Thank you for pointing this out, and yes, the experimenter was blinded to the mice genotypes during the different novel object test (and the other experiments too). We have added a comment to this point (lines 199-201, 294-295).

2. In line 425 you mention table 1, but unfortunately this table is missing.

Reply: many apologies for this oversight, it got missed off during the submission process. It has been added at the end of the manuscript (page 33).

3. Figure legends are missing. Please add this to the manuscript, as the figures are not self-explaining.

Reply: many apologies for this oversight, they got missed off during the submission process. They have been added at the end of the manuscript (pages 34-36).

4. Figure 2: To me it is unfortunately not clear what I should take out of this figure, despite the fact that some genes are upregulated, while others are downregulated. Maybe you could indicate here important genes that are altered in their expression profile, or at least the bottom-line of the dendrograms.

Reply: We understand that the unfortunate absence of the legend may have made interpretation of this figure more difficult that it should have done. However, taking on board this point, we have added further detail to the legend of this figure, indicating the numbers of genes downregulated or upregulated in our analyses. We have not highlighted key gene expression changes on the figure, as we feel this may be biased - the rigorous analysis of this is provided in Table 1. It is standard practice to include both a Volcano plot and a representation of hierarchical cluster when presenting RNA-seq data. However, we appreciate that this Figure could be moved to the Supplementary material if the Reviewer thinks this would be more appropriate. The figure legend will now read as follows:

Figure 2. Effects of Setd1a haploinsufficiency on transcriptional changes in E13.5 Setd1a+/- brain. Volcano plot (a) shows all the differentially expressed genes discovered. There were 267 genes that were differentially expressed (234 downregulated and 33 upregulated), with log2 fold changes ranging from -0.63 to 1.19. A dendrogram showing hierarchical clustering (b) indicated that samples of the same genotype clustered together, suggesting similar patterns of gene expression changes within each group.

(page 33).

5. Fig. 3: Maybe you could add the respective drugs (risperidone, haloperidol) to the figure labels.

Reply: Thank you for this suggestion to aid clarity, we have therefore labelled Fig. 3 as suggested, and also carried the format through to the other figures in the main report and the Supplementary Information (S7 Fig).

---

## [Editor Report · Decision Letter 1]

2 Jul 2024

Sex-dependent effects of Setd1a haploinsufficiency on development and adult behaviour

PONE-D-24-03923R1

Dear Dr. Humby,

We’re pleased to inform you that your manuscript has been judged scientifically suitable for publication and will be formally accepted for publication once it meets all outstanding technical requirements.

Kind regards,

Kenji Tanigaki, Ph.D., M.D.

Academic Editor

PLOS ONE

---

## [Editor Report · Acceptance letter]

4 Jul 2024

PONE-D-24-03923R1 

PLOS ONE

Dear Dr. Humby, 

I'm pleased to inform you that your manuscript has been deemed suitable for publication in PLOS ONE. Congratulations! Your manuscript is now being handed over to our production team.

Kind regards, 

on behalf of

Dr. Kenji Tanigaki 

Academic Editor

PLOS ONE